# SNAPSHOT ENSEMBLES: TRAIN 1, GET $M$ FOR FREE

**Gao Huang**[∗], **Yixuan Li,**[∗] **Geoff Pleiss**
Cornell University
{gh349, yl2363}@cornell.edu, geoff@cs.cornell.edu

**Zhuang Liu**
Tsinghua University
liuzhuangthu@gmail.com

**John E. Hopcroft, Kilian Q. Weinberger**
Cornell University
jeh@cs.cornell.edu, kqw4@cornell.edu

## ABSTRACT

Ensembles of neural networks are known to be much more robust and accurate than individual networks. However, training multiple deep networks for model averaging is computationally expensive. In this paper, we propose a method to obtain the seemingly contradictory goal of *ensembling multiple neural networks at no additional training cost*. We achieve this goal by training a single neural network, converging to several local minima along its optimization path and saving the model parameters. To obtain repeated rapid convergence, we leverage recent work on cyclic learning rate schedules. The resulting technique, which we refer to as *Snapshot Ensembling*, is simple, yet surprisingly effective. We show in a series of experiments that our approach is compatible with diverse network architectures and learning tasks. It consistently yields lower error rates than state-of-the-art single models at no additional training cost, and compares favorably with traditional network ensembles. On CIFAR-10 and CIFAR-100 our DenseNet Snapshot Ensembles obtain error rates of $3.4\%$ and $17.4\%$ respectively.

## 1 INTRODUCTION

Stochastic Gradient Descent (SGD) (Bottou, 2010) and its accelerated variants (Kingma & Ba, 2014; Duchi et al., 2011) have become the de-facto approaches for optimizing deep neural networks. The popularity of SGD can be attributed to its ability to avoid and even escape spurious saddle-points and local minima (Dauphin et al., 2014). Although avoiding these spurious solutions is generally considered positive, in this paper we argue that these local minima contain useful information that may in fact improve model performance.

Although deep networks typically never converge to a global minimum, there is a notion of "good" and "bad" local minima with respect to generalization. Keskar et al. (2016) argue that local minima with flat basins tend to generalize better. SGD tends to avoid sharper local minima because gradients are computed from small mini-batches and are therefore inexact (Keskar et al., 2016). If the learning-rate is sufficiently large, the intrinsic random motion across gradient steps prevents the optimizer from reaching any of the sharp basins along its optimization path. However, if the learning rate is small, the model tends to converge into the closest local minimum. These two very different behaviors of SGD are typically exploited in different phases of optimization (He et al., 2016a). Initially the learning rate is kept high to move into the general vicinity of a flat local minimum. Once this search has reached a stage in which no further progress is made, the learning rate is dropped (once or twice), triggering a descent, and ultimately convergence, to the final local minimum.

It is well established (Kawaguchi, 2016) that the number of possible local minima grows exponentially with the number of parameters—of which modern neural networks can have millions. It is therefore not surprising that two identical architectures optimized with different initializations or minibatch orderings will converge to different solutions. Although different local minima often have very similar error rates, the corresponding neural networks tend to make different mistakes. This

---

[∗]Authors contribute equally.

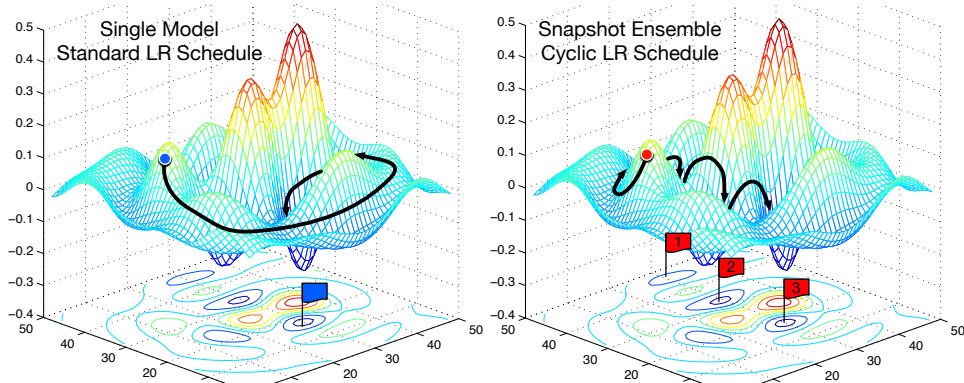

Figure 1: **Left:** Illustration of SGD optimization with a typical learning rate schedule. The model converges to a minimum at the end of training. **Right:** Illustration of Snapshot Ensembling. The model undergoes several learning rate annealing cycles, converging to and escaping from multiple local minima. We take a snapshot at each minimum for test-time ensembling.

diversity can be exploited through ensembling, in which multiple neural networks are trained from different initializations and then combined with majority voting or averaging (Caruana et al., 2004). Ensembling often leads to drastic reductions in error rates. In fact, most high profile competitions, e.g. Imagenet (Deng et al., 2009) or Kaggle[1], are won by ensembles of deep learning architectures.

Despite its obvious advantages, the use of ensembling for deep networks is not nearly as widespread as it is for other algorithms. One likely reason for this lack of adaptation may be the cost of learning multiple neural networks. Training deep networks can last for weeks, even on high performance hardware with GPU acceleration. As the training cost for ensembles increases linearly, ensembles can quickly becomes uneconomical for most researchers without access to industrial scale computational resources.

In this paper we focus on the seemingly-contradictory goal of learning an ensemble of multiple neural networks *without incurring any additional training costs*. We achieve this goal with a training method that is simple and straight-forward to implement. Our approach leverages the non-convex nature of neural networks and the ability of SGD to converge to and escape from local minima on demand. Instead of training $M$ neural networks independently from scratch, we let SGD converge $M$ times to local minima along its optimization path. Each time the model converges, we save the weights and add the corresponding network to our ensemble. We then restart the optimization with a large learning rate to escape the current local minimum. More specifically, we adopt the cycling procedure suggested by Loshchilov & Hutter (2016), in which the learning rate is abruptly raised and then quickly lowered to follow a cosine function. Because our final ensemble consists of snapshots of the optimization path, we refer to our approach as *Snapshot Ensembling*. Figure 1 presents a high-level overview of this method.

In contrast to traditional ensembles, the training time for the entire ensemble is identical to the time required to train a *single* traditional model. During testing time, one can evaluate and average the last (and therefore most accurate) $m$ out of $M$ models. Our approach is naturally compatible with other methods to improve the accuracy, such as data augmentation, stochastic depth (Huang et al., 2016b), or batch normalization (Ioffe & Szegedy, 2015). In fact, Snapshot Ensembles can even be ensembled, if for example parallel resources are available during training. In this case, an ensemble of $K$ Snapshot Ensembles yields $K \times M$ models at $K$ times the training cost.

We evaluate the efficacy of Snapshot Ensembles on three state-of-the-art deep learning architectures for object recognition: ResNet (He et al., 2016b), Wide-ResNet (Zagoruyko & Komodakis, 2016), and DenseNet (Huang et al., 2016a). We show across four different data sets that Snapshot Ensembles almost always reduce error without increasing training costs. For example, on CIFAR-10 and CIFAR-100, Snapshot Ensembles obtains error rates of $3.44\%$ and $17.41\%$ respectively.

---

[1]www.kaggle.com

## 2 RELATED WORK

Neural network ensembles have been widely studied and applied in machine learning (Hansen & Salamon, 1990; Krogh et al., 1995). However, most of these prior studies focus on improving the generalization performance, while few of them address the cost of training ensembles.

As an alternative to traditional ensembles, so-called "implicit" ensembles have high efficiency during both training and testing (Srivastava et al., 2014; Wan et al., 2013; Huang et al., 2016b; Singh et al., 2016; Krueger et al., 2016). The Dropout (Srivastava et al., 2014) technique creates an ensemble out of a single model by "dropping" — or zeroing — random sets of hidden nodes during each mini-batch. At test time, no nodes are dropped, and each node is scaled by the probability of surviving during training. Srivastava et al. claim that Dropout reduces overfitting by preventing the co-adaptation of nodes. An alternative explanation is that this mechanism creates an exponential number of networks with shared weights during training, which are then implicitly ensembled at test time. DropConnect (Wan et al., 2013) uses a similar trick to create ensembles at test time by dropping connections (weights) during training instead of nodes. The recently proposed Stochastic Depth technique (Huang et al., 2016b) randomly drops layers during training to create an implicit ensemble of networks with varying depth at test time. Finally, Swapout (Singh et al., 2016) is a stochastic training method that generalizes Dropout and Stochastic Depth. From the perspective of model ensembling, Swapout creates diversified network structures for model averaging. Our proposed method similarly trains only a single model; however, the resulting ensemble is "explicit" in that the models do not share weights. Furthermore, our method can be used in conjunction with any of these implicit ensembling techniques.

Several recent publications focus on reducing the *test time cost* of ensembles, by transferring the "knowledge" of cumbersome ensembles into a single model (Bucilu et al., 2006; Hinton et al., 2015). Hinton et al. (2015) propose to use an ensemble of multiple networks as the target of a single (smaller) network. Our proposed method is complementary to these works as we aim to reduce the *training* cost of ensembles rather than the test-time cost.

Perhaps most similar to our work is that of Swann & Allinson (1998) and Xie et al. (2013), who explore creating ensembles from slices of the learning trajectory. Xie et al. introduce the *horizontal and vertical* ensembling method, which combines the output of networks within a range of training epochs. More recently, Jean et al. (2014) and Sennrich et al. (2016) show improvement by ensembling the intermediate stages of model training. Laine & Aila (2016) propose a temporal ensembling method for semi-supervised learning, which achieves consensus among models trained with different regularization and augmentation conditions for better generalization performance. Finally, Moghimi et al. (2016) show that boosting can be applied to convolutional neural networks to create strong ensembles. Our work differs from these prior works in that we force the model to visit multiple local minima, and we take snapshots *only* when the model reaches a minimum. We believe this key insight allows us to leverage more power from our ensembles.

Our work is inspired by the recent findings of Loshchilov & Hutter (2016) and Smith (2016), who show that cyclic learning rates can be effective for training convolutional neural networks. The authors show that each cycle produces models which are (almost) competitive to those learned with traditional learning rate schedules while requiring a fraction of training iterations. Although model performance temporarily suffers when the learning rate cycle is restarted, the performance eventually surpasses the previous cycle after annealing the learning rate. The authors suggest that cycling perturbs the parameters of a converged model, which allows the model to find a better local minimum. We build upon these recent findings by (1) showing that there is significant diversity in the local minima visited during each cycle and (2) exploiting this diversity using ensembles. We are not concerned with speeding up or improving the training of a single model; rather, our goal is to extract an ensemble of classifiers while following the optimization path of the final model.

## 3 SNAPSHOT ENSEMBLING

Snapshot Ensembling produces an ensemble of accurate and diverse models from a single training process. At the heart of Snapshot Ensembling is an optimization process which visits several local minima before converging to a final solution. We take model snapshots at these various minima, and average their predictions at test time.

Ensembles work best if the individual models (1) have low test error and (2) do not overlap in the set of examples they misclassify. Along most of the optimization path, the weight assignments of a neural network tend not to correspond to low test error. In fact, it is commonly observed that the validation error drops significantly only after the learning rate has been reduced, which is typically done after several hundred epochs. Our approach is inspired by the observation that training neural networks for fewer epochs and dropping the learning rate earlier has minor impact on the final test error (Loshchilov & Hutter, 2016). This seems to suggest that local minima along the optimization path become promising (in terms of generalization error) after only a few epochs.

**Cyclic Cosine Annealing.** To converge to multiple local minima, we follow a cyclic annealing schedule as proposed by Loshchilov & Hutter (2016). We lower the learning rate at a very fast pace, encouraging the model to converge towards its first local minimum after as few as 50 epochs. The optimization is then continued at a larger learning rate, which perturbs the model and dislodges it from the minimum. We repeat this process several times to obtain multiple convergences. Formally, the learning rate $\alpha$ has the form:

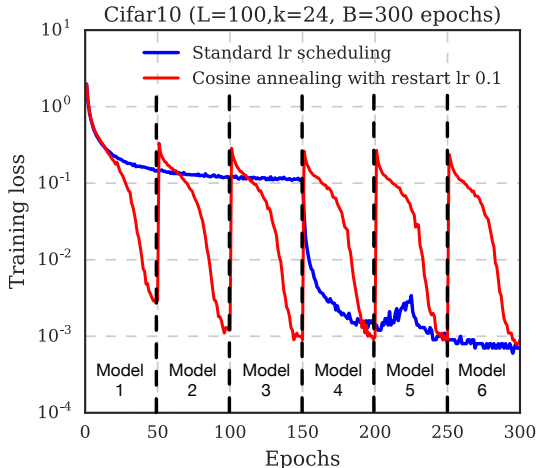

Figure 2: Training loss of 100-layer DenseNet on CI-FAR10 using standard learning rate (blue) and $M = 6$ cosine annealing cycles (red). The intermediate models, denoted by the dotted lines, form an ensemble at the end of training.

$$\alpha(t) = f\left(\mathrm{mod}\left(t - 1, \lceil T/M \rceil\right)\right), \quad (1)$$

where $t$ is the iteration number, $T$ is the total number of training iterations, and $f$ is a monotonically decreasing function. In other words, we split the training process into $M$ cycles, each of which starts with a large learning rate, which is annealed to a smaller learning rate. The large learning rate $\alpha = f(0)$ provides the model enough energy to escape from a critical point, while the small learning rate $\alpha = f(\lceil T/M \rceil)$ drives the model to a well behaved local minimum. In our experiments, we set $f$ to be the shifted cosine function proposed by Loshchilov & Hutter (2016):

$$\alpha(t) = \frac{\alpha_0}{2} \left( \cos\left( \frac{\pi \mathrm{mod}(t - 1, \lceil T/M \rceil)}{\lceil T/M \rceil} \right) + 1 \right), \quad (2)$$

where $\alpha_0$ is the initial learning rate. Intuitively, this function anneals the learning rate from its initial value $\alpha_0$ to $f(\lceil T/M \rceil) \approx 0$ over the course of a cycle. Following (Loshchilov & Hutter, 2016), we update the learning rate at each iteration rather than at every epoch. This improves the convergence of short cycles, even when a large initial learning rate is used.

**Snapshot Ensembling.** Figure 2 depicts the training process using cyclic and traditional learning rate schedules. At the end of each training cycle, it is apparent that the model reaches a local minimum with respect to the training loss. Thus, before raising the learning rate, we take a "snapshot" of the model weights (indicated as vertical dashed black lines). After training $M$ cycles, we have $M$ model snapshots, $f_1 \ldots f_M$, each of which will be used in the final ensemble. It is important to highlight that the total training time of the $M$ snapshots is the same as training a model with a standard schedule (indicated in blue). In some cases, the standard learning rate schedule achieves lower training loss than the cyclic schedule; however, as we will show in the next section, the benefits of ensembling outweigh this difference.

**Ensembling at Test Time.** The ensemble prediction at test time is the average of the last $m$ ($m \leq M$) model's softmax outputs. Let $\mathbf{x}$ be a test sample and let $h_i(\mathbf{x})$ be the softmax score of snapshot $i$. The output of the ensemble is a simple average of the last $m$ models: $h_{\mathrm{Ensemble}} = \frac{1}{m} \sum_0^{m-1} h_{M-i}(\mathbf{x})$. We always ensemble the last $m$ models, as these models tend to have the lowest test error.

| | Method | C10 | C100 | SVHN | Tiny ImageNet |
|---|---|---|---|---|---|
| ResNet-110 | Single model | 5.52 | 28.02 | 1.96 | 46.50 |
| | NoCycle Snapshot Ensemble | 5.49 | 26.97 | 1.78 | 43.69 |
| | SingleCycle Ensembles | 6.66 | 24.54 | 1.74 | 42.60 |
| | Snapshot Ensemble ($\alpha_0 = 0.1$) | 5.73 | 25.55 | **1.63** | 40.54 |
| | Snapshot Ensemble ($\alpha_0 = 0.2$) | **5.32** | **24.19** | 1.66 | **39.40** |
| Wide-ResNet-32 | Single model | 5.43 | 23.55 | 1.90 | 39.63 |
| | Dropout | 4.68 | 22.82 | 1.81 | 36.58 |
| | NoCycle Snapshot Ensemble | 5.18 | 22.81 | 1.81 | 38.64 |
| | SingleCycle Ensembles | 5.95 | 21.38 | 1.65 | 35.53 |
| | Snapshot Ensemble ($\alpha_0 = 0.1$) | **4.41** | **21.26** | 1.64 | 35.45 |
| | Snapshot Ensemble ($\alpha_0 = 0.2$) | 4.73 | 21.56 | **1.51** | **32.90** |
| DenseNet-40 | Single model | 5.24* | 24.42* | 1.77 | 39.09 |
| | Dropout | 6.08 | 25.79 | 1.79* | 39.68 |
| | NoCycle Snapshot Ensemble | 5.20 | 24.63 | 1.80 | 38.51 |
| | SingleCycle Ensembles | 5.43 | 22.51 | 1.87 | 38.00 |
| | Snapshot Ensemble ($\alpha_0 = 0.1$) | 4.99 | 23.34 | **1.64** | 37.25 |
| | Snapshot Ensemble ($\alpha_0 = 0.2$) | **4.84** | **21.93** | 1.73 | **36.61** |
| DenseNet-100 | Single model | 3.74* | 19.25* | - | - |
| | Dropout | 3.65 | 18.77 | - | - |
| | NoCycle Snapshot Ensemble | 3.80 | 19.30 | - | - |
| | SingleCycle Ensembles | 4.52 | 18.38 | | |
| | Snapshot Ensemble ($\alpha_0 = 0.1$) | 3.57 | 18.12 | - | - |
| | Snapshot Ensemble ($\alpha_0 = 0.2$) | **3.44** | **17.41** | - | - |

Table 1: Error rates (%) on CIFAR-10 and CIFAR-100 datasets. All methods in the same group are trained for the same number of iterations. Results of our method are colored in blue, and the best result for each network/dataset pair are **bolded**. * indicates numbers which we take directly from Huang et al. (2016a).

## 4 EXPERIMENTS

We demonstrate the effectiveness of Snapshot Ensembles on several benchmark datasets, comparing with competitive baselines. We run all experiments with Torch 7 (Collobert et al., 2011)[2].

### 4.1 DATASETS

**CIFAR.** The two CIFAR datasets (Krizhevsky & Hinton, 2009) consist of colored natural images sized at 32×32 pixels. CIFAR-10 (C10) and CIFAR-100 (C100) images are drawn from 10 and 100 classes, respectively. For each dataset, there are 50,000 training images and 10,000 images reserved for testing. We use a standard data augmentation scheme (Lin et al., 2013; Romero et al., 2014; Lee et al., 2015; Springenberg et al., 2014; Srivastava et al., 2015; Huang et al., 2016b; Larsson et al., 2016), in which the images are zero-padded with 4 pixels on each side, randomly cropped to produce 32×32 images, and horizontally mirrored with probability 0.5.

**SVHN.** The Street View House Numbers (SVHN) dataset (Netzer et al., 2011) contains $32 \times 32$ colored digit images from Google Street View, with one class for each digit. There are 73,257 images in the training set and 26,032 images in the test set. Following common practice (Sermanet et al., 2012; Goodfellow et al., 2013; Huang et al., 2016a), we withhold 6,000 training images for validation, and train on the remaining images without data augmentation.

**Tiny ImageNet.** The Tiny ImageNet dataset[3] consists of a subset of ImageNet images (Deng et al., 2009). There are 200 classes, each of which has 500 training images and 50 validation images. Each image is resized to $64 \times 64$ and augmented with random crops, horizontal mirroring, and RGB intensity scaling (Krizhevsky et al., 2012).

**ImageNet.** The ILSVRC 2012 classification dataset (Deng et al., 2009) consists of 1000 images classes, with a total of 1.2 million training images and 50,000 validation images. We adopt the same

---

[2]Code to reproduce results is available at `https://github.com/gaohuang/SnapshotEnsemble`
[3]`https://tiny-imagenet.herokuapp.com`

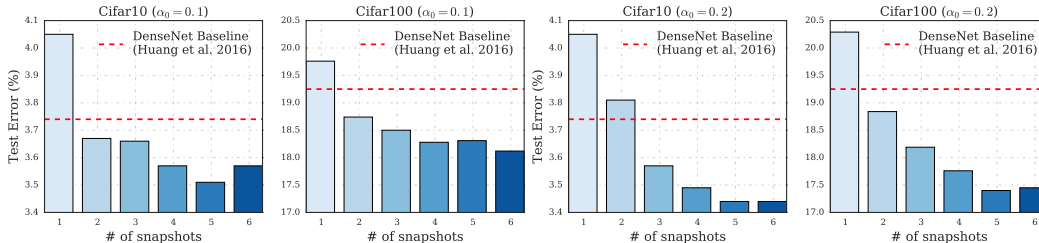

Figure 3: DenseNet-100 Snapshot Ensemble performance on CIFAR-10 and CIFAR-100 with restart learning rate $\alpha_0 = 0.1$ (left two) and $\alpha_0 = 0.2$ (right two). Each ensemble is trained with $M = 6$ annealing cycles (50 epochs per each).

data augmentation scheme as in (He et al., 2016a; Huang et al., 2016a) and apply a $224 \times 224$ center crop to images at test time.

## 4.2 TRAINING SETTING

**Architectures.** We test several state-of-the-art architectures, including residual networks (**ResNet**) (He et al., 2016a), **Wide ResNet** (Zagoruyko & Komodakis, 2016) and **DenseNet** (Huang et al., 2016a). For ResNet, we use the original 110-layer network introduced by He et al. (2016a). Wide-ResNet is a 32-layer ResNet with 4 times as many convolutional features per layer as a standard ResNet. For DenseNet, our large model follows the same setup as (Huang et al., 2016a), with depth $L = 100$ and growth rate $k = 24$. In addition, we also evaluate our method on a small DenseNet, with depth $L = 40$ and $k = 12$. To adapt all these networks to Tiny ImageNet, we add a stride of 2 to the first layer of the models, which downsamples the images to $32 \times 32$. For ImageNet, we test the 50-layer ResNet proposed in (He et al., 2016a). We use a mini batch size of 64.[4]

**Baselines.** Snapshot Ensembles incur the training cost of a single model; therefore, we compare with baselines that require the same amount of training. First, we compare against a **Single Model** trained with a standard learning rate schedule, dropping the learning rate from 0.1 to 0.01 halfway through training, and then to 0.001 when training is at 75%. Additionally, to compare against implicit ensembling methods, we test against a single model trained with **Dropout**. This baseline uses the same learning rate as above, and drops nodes during training with a probability of 0.2.

We then test the **Snapshot Ensemble** algorithm trained with the cyclic cosine learning rate as described in (2). We test models with the max learning rate $\alpha_0$ set to 0.1 and 0.2. In both cases, we divide the training process into learning rate cycles. Model snapshots are taken after each learning rate cycle. Additionally, we train a Snapshot Ensemble with a non-cyclic learning rate schedule. This **NoCycle Snapshot Ensemble**, which uses the same schedule as the Single Model and Dropout baselines, is meant to highlight the impact of cyclic learning rates for our method. To accurately compare with the cyclic Snapshot Ensembles, we take the same number of snapshots equally spaced throughout the training process. Finally, we compare against **SingleCycle Ensembles**, a Snapshot Ensemble variant in which the network is re-initialized at the beginning of every cosine learning rate cycle, rather than using the parameters from the previous optimization cycle. This baseline essentially creates a traditional ensemble, yet each network only has $1/M$ of the typical training time. This variant is meant to highlight the tradeoff between model diversity and model convergence. Though SingleCycle Ensembles should in theory explore more of the parameter space, the models do not benefit from the optimization of previous cycles.

**Training Budget.** On CIFAR datasets, the training budget is $B = 300$ epochs for DenseNet-40 and DenseNet-100, and $B = 200$ for ResNet and Wide ResNet models. Snapshot variants are trained with $M = 6$ cycles of $B/M = 50$ epochs for DenseNets, and $M = 5$ cycles of $B/M = 40$ epochs for ResNets/Wide ResNets. SVHN models are trained with a budget of $B = 40$ epochs (5 cycles of 8 epochs). For Tiny ImageNet, we use a training budget of $B = 150$ (6 cycles of 25 epochs). Finally, ImageNet is trained with a budget of $B = 90$ epochs, and we trained 2 Snapshot variants: one with $M = 2$ cycles and one with $M = 3$.

---

[4]Exceptions: ResNet-110 and Wide-ResNet are trained with batch size 128 on Tiny ImageNet. The ImageNet model is trained with batch size 256.

## 4.3 SNAPSHOT ENSEMBLE RESULTS

**Accuracy.** The main results are summarized in Table 1. In most cases, Snapshot ensembles achieve lower error than any of the baseline methods. Most notably, Snapshot Ensembles yield an error rate of $17.41\%$ on CIFAR-100 using large DenseNets, far outperforming the record of $19.25\%$ under the same training cost and architecture (Huang et al., 2016a). Our method has the most success on CIFAR-100 and Tiny ImageNet, which is likely due to the

| Method | Val. Error (%) |
|---|---|
| Single model | 24.01 |
| Snapshot Ensemble ($M = 2$) | 23.33 |
| Snapshot Ensemble ($M = 3$) | 23.96 |

Table 2: Top-1 error rates (%) on ImageNet validation set using ResNet-50 with varying number of cycles.

complexity of these datasets. The softmax outputs for these datasets are high dimensional due to the large number of classes, making it unlikely that any two models make the same predictions. Snapshot Ensembling is also capable of improving the competitive baselines for CIFAR-10 and SVHN as well, reducing error by $1\%$ and $0.4\%$ respectively with the Wide ResNet architecture.

The NoCycle Snapshot Ensemble generally has little effect on performance, and in some instances even *increases* the test error. This highlights the need for a cyclic learning rate for useful ensembling. The SingleCycle Ensemble has similarly mixed performance. In some cases, e.g., DenseNet-40 on CIFAR-100, the SingleCycle Ensemble is competitive with Snapshot Ensembles. However, as the model size increases to 100 layers, it does not perform as well. This is because it is difficult to train a large model from scratch in only a few epochs. These results demonstrate that Snapshot Ensembles tend to work best when utilizing information from previous cycles. Effectively, Snapshot Ensembles strike a balance between model diversity and optimization.

Table 2 shows Snapshot Ensemble results on ImageNet. The Snapshot Ensemble with $M = 2$ achieves $23.33\%$ validation error, outperforming the single model baseline with $24.01\%$ validation error. It appears that 2 cycles is the optimal choice for the ImageNet dataset. Provided with the limited total training budget $B = 90$ epochs, we hypothesize that allocating fewer than $B/2 = 45$ epochs per training cycle is insufficient for the model to converge on such a large dataset.

**Ensemble Size.** In some applications, it may be beneficial to vary the size of the ensemble *dynamically* at test time depending on available resources. Figure 3 displays the performance of DenseNet-40 on the CIFAR-100 dataset as the effective ensemble size, $m$, is varied. Each ensemble consists of snapshots from later cycles, as these snapshots have received the most training and therefore have likely converged to better minima. Although ensembling more models generally gives better performance, we observe significant drops in error when the second and third models are added to the ensemble. In most cases, an ensemble of two models outperforms the baseline model.

| $M$ | Test Error (%) |
|---|---|
| 2 | 22.92 |
| 4 | 22.07 |
| 6 | 21.93 |
| 8 | 21.89 |
| 10 | 22.16 |

Table 3: Error rates of a DenseNet-40 Snapshot Ensemble on CIFAR-100, varying $M$—the number of models (cycles) used in the ensemble.

**Restart Learning Rate.** The effect of the restart learning rate can be observed in Figure 3. The left two plots show performance when using a restart learning rate of $\alpha_0 = 0.1$ at the beginning of each cycle, and the right two plots show $\alpha_0 = 0.2$. In most cases, ensembles with the larger restart learning rate perform better, presumably because the strong perturbation in between cycles increases the diversity of local minima.

**Varying Number of Cycles.** Given a fixed training budget, there is a trade-off between the number of learning rate cycles and their length. Therefore, we investigate how the number of cycles $M$ affects the ensemble performance, given a fixed training budget. We train a 40-layer DenseNet on the CIFAR-100 dataset with an initial learning rate of $\alpha_0 = 0.2$. We fix the total training budget $B = 300$ epochs, and vary the value of $M \in \{2, 4, 6, 8, 10\}$. As shown in Table 3, our method is relatively robust with respect to different values of $M$. At the extremes, $M = 2$ and $M = 10$, we find a slight degradation in performance, as the cycles are either too few or too short. In practice, we find that setting $M$ to be $4 \sim 8$ works reasonably well.

**Varying Training Budget.** The left and middle panels of Figure 4 show the performance of Snapshot Ensembles and SingleCycle Ensembles as a function of training budget (where the number of cycles is fixed at $M = 6$). We train a 40-layer DenseNet on CIFAR-10 and CIFAR-100, with an initial learning rate of $\alpha_0 = 0.1$, varying the total number of training epochs from 60 to 300. We observe

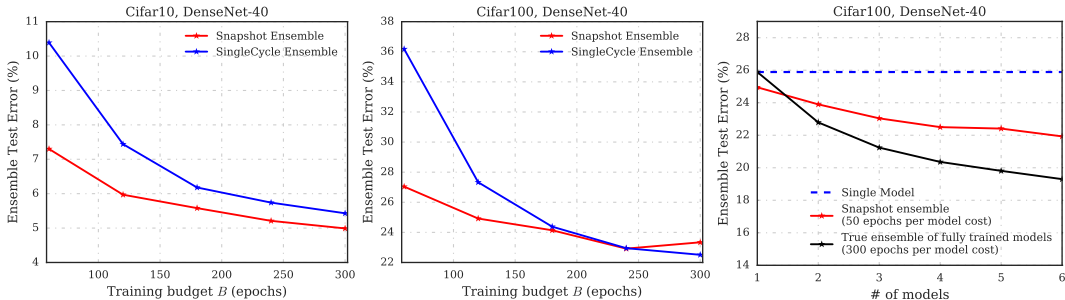

Figure 4: Snapshot Ensembles under different training budgets on (**Left**) CIFAR-10 and (**Middle**) CIFAR-100. **Right:** Comparison of Snapshot Ensembles with true ensembles.

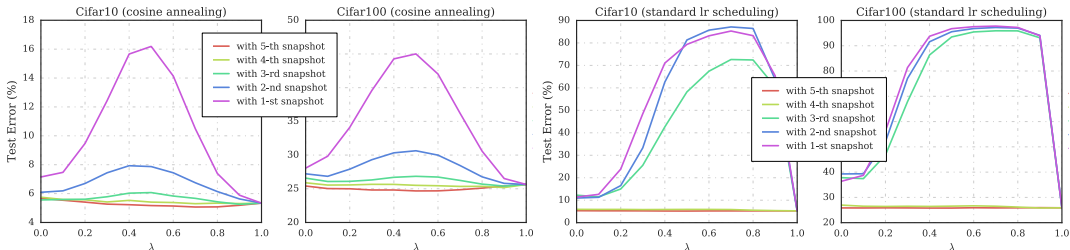

Figure 5: Interpolations in parameter space between the final model (sixth snapshot) and all intermediate snapshots. $\lambda = 0$ represents an intermediate snapshot model, while $\lambda = 1$ represents the final model. **Left:** A Snapshot Ensemble, with cosine annealing cycles ($\alpha_0 = 0.2$ every $B/M = 50$ epochs). **Right:** A NoCycle Snapshot Ensemble, (two learning rate drops, snapshots every 50 epochs).

that both Snapshot Ensembles and SingleCycle Ensembles become more accurate as training budget increases. However, we note that as training budget decreases, Snapshot Ensembles still yield competitive results, while the performance of the SingleCycle Ensembles degrades rapidly. These results highlight the improvements that Snapshot Ensembles obtain when the budget is low. If the budget is high, then the SingleCycle baseline approaches true ensembles and outperforms Snapshot ensembles eventually.

**Comparison with True Ensembles.** We compare Snapshot Ensembles with the traditional ensembling method. The right panel of Figure 4 shows the test error rates of DenseNet-40 on CIFAR-100. The true ensemble method averages models that are trained with 300 full epochs, each with different weight initializations. Given the same number of models at test time, the error rate of the true ensemble can be seen as a lower bound of our method. Our method achieves performance that is comparable with ensembling of 2 independent models, but with the training cost of one model.

## 4.4 DIVERSITY OF MODEL ENSEMBLES

**Parameter Space.** We hypothesize that the cyclic learning rate schedule creates snapshots which are not only accurate but also diverse with respect to model predictions. We qualitatively measure this diversity by visualizing the local minima that models converge to. To do so, we linearly interpolate snapshot models, as described by Goodfellow et al. (2014). Let $J(\theta)$ be the test error of a model using parameters $\theta$. Given $\theta_1$ and $\theta_2$ — the parameters from models 1 and 2 respectively — we can compute the loss for a convex combination of model parameters: $J(\lambda(\theta_1) + (1 - \lambda)(\theta_2))$, where $\lambda$ is a mixing coefficient. Setting $\lambda$ to 1 results in a parameters that are entirely $\theta_1$ while setting $\lambda$ to 0 gives the parameters $\theta_2$. By sweeping the values of $\lambda$, we can examine a linear slice of the parameter space. Two models that converge to a similar minimum will have smooth parameter interpolations, whereas models that converge to different minima will likely have a non-convex interpolation, with a spike in error when $\lambda$ is between 0 and 1.

Figure 5 displays interpolations between the final model of DenseNet-40 (sixth snapshot) and all intermediate snapshots. The left two plots show Snapshot Ensemble models trained with a cyclic learning rate, while the right two plots show NoCycle Snapshot models. $\lambda = 0$ represents a model which is entirely snapshot parameters, while $\lambda = 1$ represents a model which is entirely the parameters of the final model. From this figure, it is clear that there are differences between cyclic and

non-cyclic learning rate schedules. Firstly, all of the cyclic snapshots achieve roughly the same error as the final cyclical model, as the error is similar for $\lambda = 0$ and $\lambda = 1$. Additionally, it appears that most snapshots do not lie in the same minimum as the final model. Thus the snapshots are likely to misclassify different samples. Conversely, the first three snapshots achieve much higher error than the final model. This can be observed by the sharp minima around $\lambda = 1$, which suggests that mixing in any amount of the snapshot parameters will worsen performance. While the final two snapshots achieve low error, the figures suggests that they lie in the same minimum as the final model, and therefore likely add limited diversity to the ensemble.

**Activation space.** To further explore the diversity of models, we compute the pairwise correlation of softmax outputs for every pair of snapshots. Figure 6 displays the average correlation for both cyclic snapshots and non-cyclical snapshots. Firstly, there are large correlations between the last 3 snapshots of the non-cyclic training schedule (right). These snapshots are taken after dropping the learning rate, suggesting that each snapshot has converged to the same minimum. Though there is more diversity amongst the earlier snapshots, these snapshots have much higher error rates and are therefore not ideal for ensembling. Conversely, there is less correlation between all cyclic snapshots (left). Because all snapshots have similar accuracy (as can be seen in Figure 5), these differences in predictions can be exploited to create effective ensembles.

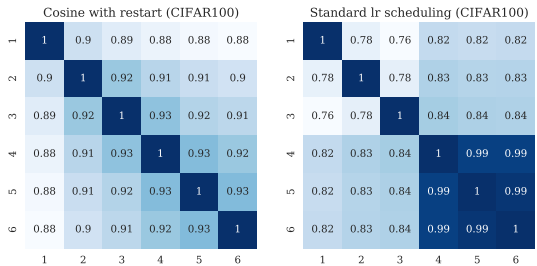

Figure 6: Pairwise correlation of softmax outputs between any two snapshots for DenseNet-100. **Left:** A Snapshot Ensemble, with cosine annealing cycles (restart with $\alpha_0 = 0.2$ every 50 epochs). **Right:** A NoCycle Snapshot Ensemble, (two learning rate drops, snapshots every 50 epochs).

## 5 DISCUSSION

We introduce Snapshot Ensembling, a simple method to obtain ensembles of neural networks without any additional training cost. Our method exploits the ability of SGD to converge to and escape from local minima as the learning rate is lowered, which allows the model to visit several weight assignments that lead to increasingly accurate predictions over the course of training. We harness this power with the cyclical learning rate schedule proposed by Loshchilov & Hutter (2016), saving model snapshots at each point of convergence. We show in several experiments that all snapshots are accurate, yet produce different predictions from one another, and therefore are well suited for test-time ensembles. Ensembles of these snapshots significantly improve the state-of-the-art on CIFAR-10, CIFAR-100 and SVHN. Future work will explore combining Snapshot Ensembles with traditional ensembles. In particular, we will investigate how to balance growing an ensemble with new models (with random initializations) and refining existing models with further training cycles under a fixed training budget.

## ACKNOWLEDGEMENTS

We thank Ilya Loshchilov and Frank Hutter for their insightful comments on the cyclic cosine-shaped learning rate. The authors are supported in part by the, III-1618134, III-1526012, IIS-1149882 grants from the National Science Foundation, US Army Research Office W911NF-14-1-0477, and the Bill and Melinda Gates Foundation.

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

SUPPLEMENTARY

**A. Single model and Snapshot Ensemble performance over time**

In Figures 7-9, we compare the test error of Snapshot Ensembles with the error of individual model snapshots. The blue curve shows the test error of a single model snapshot using a cyclic cosine learning rate. The green curve shows the test error when ensembling model snapshots over time. (Note that, unlike Figure 3, we construct these ensembles beginning with the earliest snapshots.) As a reference, the red dashed line in each panel represents the test error of single model trained for 300 epochs using a standard learning rate schedule. Without Snapshot Ensembles, in about half of the cases, the test error of final model using a cyclic learning rate—the right most point in the blue curve—is no better than using a standard learning rate schedule.

One can observe that under almost all settings, complete Snapshot Ensembles—the right most points of the green curves—outperform the single model baselines. In many cases, ensembles of just 2 or 3 model snapshots are able to match the performance of the single model trained with a standard learning rate. Not surprisingly, the ensembles of model snapshots consistently outperform any of its members, yielding a smooth curve of test error over time.

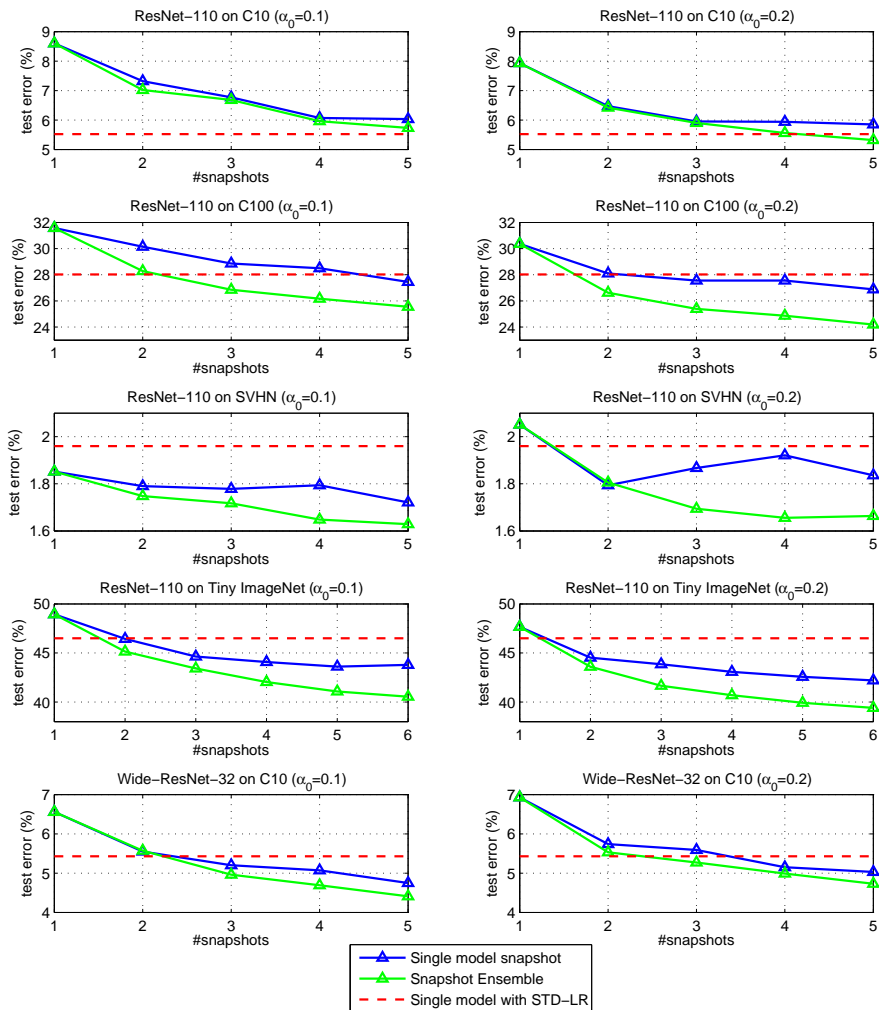

Figure 7: Single model and Snapshot Ensemble performance over time (part 1).

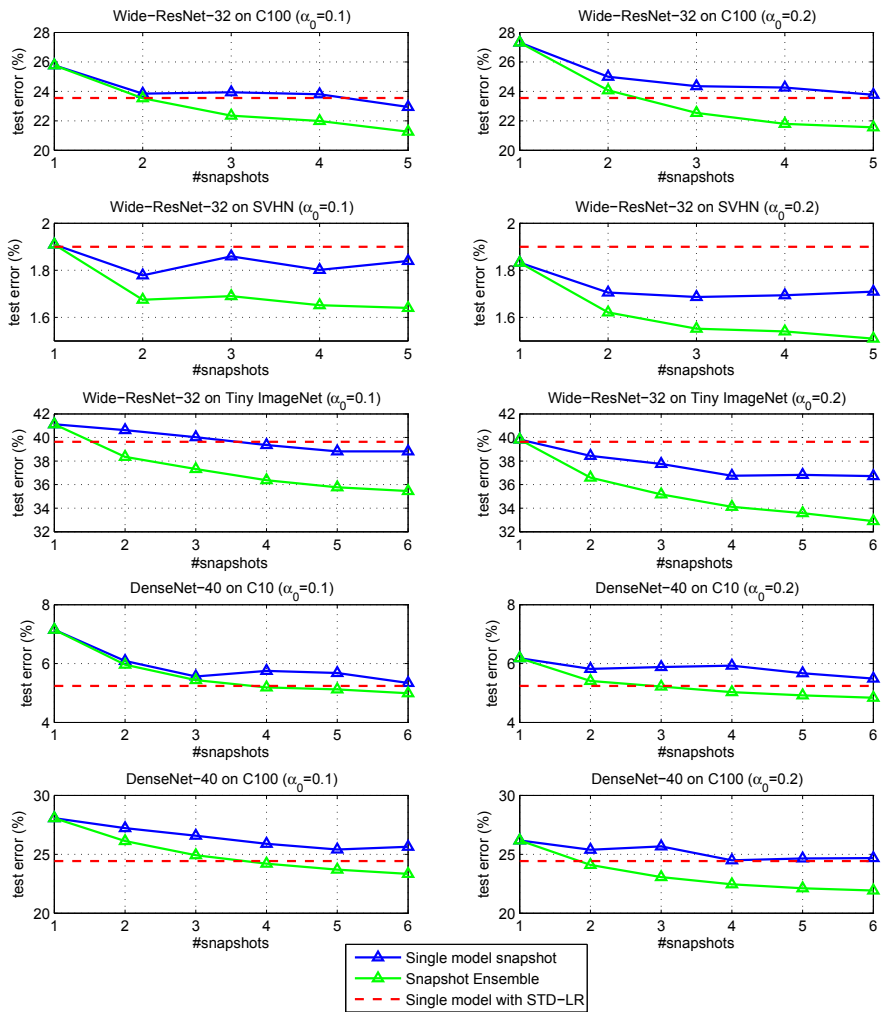

Figure 8: Single model and Snapshot Ensemble performance over time (part 2).

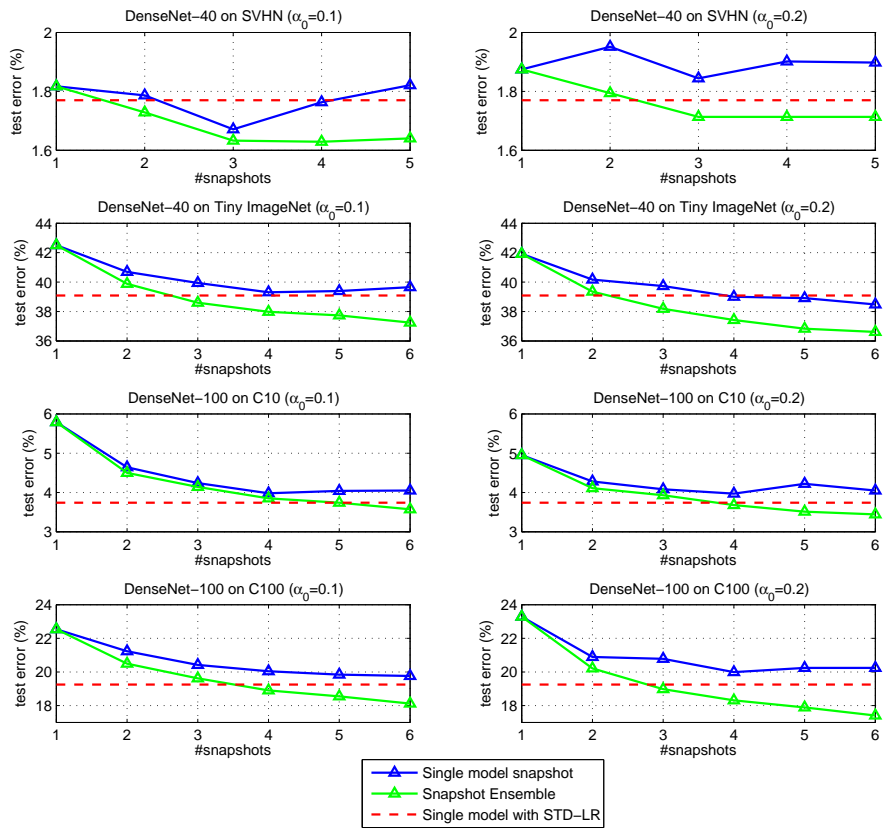

Figure 9: Single model and Snapshot Ensemble performance over time (part 3).

