# Peer review of "Snapshot Ensembles: Train 1, Get M for Free"

_ICLR 2017 — accepted_

[Public Comment · Dmytro Mishkin · 07 Nov 2016]
**Missing reference, which is extremely relevant**

Horizontal and Vertical Ensemble with Deep Representation for Classification
Jingjing Xie, Bing Xu, Zhang Chuang

page 3: "Similar to the horizontal voting method in section 3.2, it takes the output of networks within a continuous range of epoch"

[Public Comment · (anonymous) · 07 Nov 2016]
**Comparison with Ensemble method**

This paper presents an interesting take on getting an ensemble for free whilst training a single network. However, your main accuracy comparison seems to exclude traditional ensemble methods, save for the very end of section 4.3, where the actual ensemble method you used is not mentioned. I would advise expanding this paragraph to explain what ensemble technique you compared against.

Did you try comparing your results with other traditional ensembles like Bagging or AdaBoost? What were your results there? I understand that of course these are more expensive to train, but it would provide a baseline for a fair comparison of your results. It would be very interesting if you got close (or even beat!) these traditional expensive methods. At which point a comparison of training times would also be useful.

[Official Review · AnonReviewer3 · rating 9 · confidence 4 · 16 Dec 2016]

This work develops a method to quickly produce an ensemble of deep networks that outperform a single network trained for an equivalent amount of time. The basis of this approach is to use a cyclic learning rate to quickly settle the model into a local minima and saving a model snapshot at this time before quickly raising the learning rate to escape towards a different minima's well of attraction. The resulting snapshots can be collected throughout a single training run and achieve reasonable performance compared to baselines and have some of the gains of traditional ensembles (at a much lower cost). 

This paper is well written, has clear and informative figures/tables, and provides convincing results across a broad range of models and datasets. I especially liked the analysis in Section 4.4.  The publicly available code to ensure reproducibility is also greatly appreciated.

I would like to see more discussion of the accuracy and variability of each snapshot and further comparison with true ensembles.

Preliminary rating:
This is an interesting work with convincing experiments and clear writing. 

Minor note:
Why is the axis for lambda from -1 to 2 in Figure 5 where lambda is naturally between 0 and 1.

[Official Review · AnonReviewer2 · rating 7 · confidence 5 · 16 Dec 2016]
**A good paper.**
soundness 4

I don't have much to add to my pre-review questions. The main thing I'd like to see that would strengthen my review further is a larger scale evaluation, more discussion of the hyperparameters, etc. Where test error are reported for snapshot ensembles it would be useful to report statistics about the performance of individual ensemble members for comparison (mean and standard deviation, maybe best single member's error rate).

[Official Review · AnonReviewer1 · rating 8 · confidence 3 · 16 Dec 2016]
soundness 5 · clarity 4 · substance 3

The work presented in this paper proposes a method to get an ensemble of neural networks at no extra training cost (i.e., at the cost of training a single network), by saving snapshots of the network during training. Network is trained using a cyclic (cosine) learning rate schedule; the snapshots are obtained when the learning rate is at the lowest points of the cycles. Using these snapshot ensembles, they show gains in performance over a single network on the image classification task on a variety of datasets.


Positives:

1. The work should be easy to adopt and re-produce, given the simple techinque and the experimental details in the paper.
2. Well written paper, with clear description of the method and thorough experiments.


Suggestions for improvement / other comments:

1. While it is fair to compare against other techniques assuming a fixed computational budget, for a clear perspective, thorough comaprisons with "true ensembles" (i.e., ensembles of networks trained independently) should be provided.
Specificially, Table 4 should be augmented with results from "true ensembles".

2. Comparison with true ensembles is only provided for DenseNet-40 on CIFAR100 in Figure 4. The proposed snapshot ensemble achieves approximately 66% of the improvement of "true ensemble" over the single baseline model. This is not reflected accurately in the authors' claim in the abstract: "[snapshot ensembles] **almost match[es]** the results of far more expensive independently trained [true ensembles]."

3. As mentioned before: to understand the diversity of snapshot ensembles, it would help to the diversity against different ensembling technique, e.g. (1) "true ensembles", (2) ensembles from dropout as described by Gal et. al, 2016 (Dropout as a Bayesian Approximation).

[Public Comment · Bing Xu · 18 Dec 2016]
**Difference to Horizontal Ensemble?**

Congratulation to the authors for getting very high review score. However I want to know the difference between Snapshot Ensembles and Horizontal Ensemble (Thank Dmytro Mishkin for reading my undergrad paper).

I take a quick look of the paper, seems Snapshot Ensembles is very similar to Horizontal Ensemble, maybe with a different LR schedule?

[Public Comment · Gavin Brown · 27 Dec 2016]
**Related work**

There is a related paper to this work - "Fast Committee Learning", by Swann and Allinson, 1998.

[Public Comment · George Philipp · 06 Jan 2017]
**Questions**

After reading the paper, it appears to have a fatal flaw. In addition, it seems to me to make very disingenuous claims about its own experimental results.

As far as I can tell, this is how the authors train their snapshot ensemble:

(1) initialize the model
(2) train for a short period (e.g. 50 epochs for DenseNet) with a cosine schedule
(3) take a snapshot
(4) continue training with a "new" cosine schedule for another short period
(5) take a snapshot
(6) repeat steps 4 and 5 a few times
(7) ensemble the snapshots

Call this strategy A

Then the authors also discuss another strategy they call "full ensembling", which they compare to in Figure 4:

(1) initialize the model
(2) train for a long period (e.g. 300 epochs) with a "traditional" schedule
(3) take a snapshot
(4) re-initialize the model randomly
(5) train for a long period (e.g. 300 epochs) with a "traditional" schedule
(6) take a snapshot
(7) repeat steps (4) through (6) a few times
(8) ensemble the snapshots

Call this strategy B

However, consider the following strategy C:

(1) initialize the model
(2) train for a short period (e.g. 50 epochs in Figure 2) with a cosine schedule
(3) take a snapshot
(4) re-initialize the model randomly
(5) train for a short period (e.g. 50 epochs in Figure 2) with a cosine schedule
(6) take a snapshot
(7) repeat steps (4) through (6) a few times
(8) ensemble the snapshots

Strategy C is likely superior to strategy A, and this seems to invalidates the entire claim of the paper that it we should build ensembles by taking snapshots along the training path. It is likely better to train many independent models each with the cosine schedule for a short period instead of taking snapshots. Therefore, the paper appears to be a de-novation compared to full ensembling, not an innovation. This is obscured by the fact that the authors only compare to strategy B, which is a very weak baseline. Of course, if I want to train an ensemble quickly, it seems suboptimal to train any individual model for a very long time, which strategy B does.

The reason I believe that strategy C is likely superior to strategy A is two-fold. First, in Figure 3 you show that beginning the cosine schedule with a higher step size (and thus more re-randomization) leads to better performance as the ensemble grows. Therefore, total re-randomization likely does even better. Further, in Figure 4, you show that full ensembling enjoys a better slope of improvement as the ensemble grows. Strategy C also likely enjoys this slope, whereas strategy A does not.

However, even if it turns out that Strategy A indeed outperforms strategy C, not showing this in the paper I would still consider a fatal flaw unless such results were to be included. To be precise, what I would like to see is a comparison between an ensemble of the FIRST k snapshots following your strategy (strategy A), and k independently trained models that were each trained under the the cosine schedule (strategy C), for various values of k.

There is one use case where I could possibly imagine using strategy A: If I have time to train roughly for the length of one traditional schedule and want an ensemble that contains one model that has, by itself, a good performance. In that case, strategy A would outperform strategy B on the ensemble and strategy C on the single model, though A would lose to B on the single model and to C on the ensemble. In that specific case, we cannot say that there is a strategy that dominates strategy A. However, this claim does not seem strong enough as the main contribution of a top conference paper.

Looking at the high scores from the three reviewers makes me think I somehow missed something in my analysis. I am willing to change my opinion on this. Please enlighten me.

.
.
.

Another thing I want to point out is that the authors seem to make very disingenuous claims about their experimental results in at least 3 instances:

(1) At the end of section 4.3 the authors state "Our method achieves with comparable performance of ensembling 3 independant models, but with the training cost of one model." However, in Figure 4 we find that the snapshot ensemble is halfway between ensembling 2 and 3 independent models. This is not the same thing as "comparable to 3 independent models". Further, the difference in test error between the snapshot ensemble and 3 independent models is almost 1 per cent. I would not call such a difference comparable. Of course, since the baseline is extremely weak as I mentioned above, the graph has little value to begin with.

(2)  In the abstract, the authors state "It [snapshot ensembling] consistently yields significantly lower error rates than state-of-the-art single models at no additional training cost, and almost matches the results of (far more expensive) independently trained network ensembles." Looking at Figure 4, you don't come close!

(3) The authors repeatedly claim to get "multiple neural network at no additional training cost". However then they point out "It is also fair to point out that the training loss under the standard schedule is eventually lower". So they admit that if the stagewise cosine schedule takes M epochs to train (say 300), then the standard schedule achieves a lower error in those same M epochs. Therefore, the standard schedule attains the same error as the stagewise cosine schedule at some earlier epoch, say N. Therefore, the authors DO incur an additional cost in obtaining their ensemble, and it is M - N epochs.

.
.
.

One final question: The authors state that "It is also fair to point out that the training loss under the standard schedule is eventually lower". However in Figure 4 it appears that a snapshot ensemble of size 1 (i.e. the final error under the stagewise cosine schedule) has lower error than a full ensemble of size 1, i.e. the final error under the traditional schedule. This seems to be a contradiction.

.
.
.

I do agree that the general motivation of the paper is sound and the paper is, apart from the issues discussed above, well written. Again, I am inviting the authors and everybody else to prove me wrong on all of my points.

[Final Decision · Program Chairs · 06 Feb 2017]
**ICLR committee final decision**

Significant problem, interesting and simple solution, broad evaluation, authors highly responsive in incorporating feedback, all reviewers recommend acceptance. I agree.